# Vehicle Target Detection Method for Wide-Area SAR Images Based on Coarse-Grained Judgment and Fine-Grained Detection

Yucheng Song [1], Shuo Wang [2], Qing Li [1], Hongbin Mu [2], Ruyi Feng [3], Tian Tian [1,*] and Jinwen Tian [1]

[1] School of Artificial Intelligence and Automation, Huazhong University of Science and Technology, Wuhan 430074, China; yuchengsong@hust.edu.cn (Y.S.); m201972573@hust.edu.cn (Q.L.); jwtian@mail.hust.edu.cn (J.T.)
[2] Beijing Institute of Astronautical Systems Engineering, Beijing 100076, China; wanhshuo-hit@163.com (S.W.); muhongbinbit@126.com (H.M.)
[3] School of Computer Science, China University of Geosciences, Wuhan 430074, China; fengry@cug.edu.cn
* Correspondence: ttian@hust.edu.cn

**Abstract:** The detection of vehicle targets in wide-area Synthetic Aperture Radar (SAR) images is crucial for real-time reconnaissance tasks and the widespread application of remote sensing technology in military and civilian fields. However, existing detection methods often face difficulties in handling large-scale images and achieving high accuracy. In this study, we address the challenges of detecting vehicle targets in wide-area SAR images and propose a novel method that combines coarse-grained judgment with fine-grained detection to overcome these challenges. Our proposed vehicle detection model is based on YOLOv5, featuring a CAM attention module, CAM-FPN network, and decoupled detection head, and it is strengthened with background-assisted supervision and coarse-grained judgment. These various techniques not only improve the accuracy of the detection algorithms, but also enhance SAR image processing speed. We evaluate the performance of our model using the Wide-area SAR Vehicle Detection (WSVD) dataset. The results demonstrate that the proposed method achieves a high level of accuracy in identifying vehicle targets in wide-area SAR images. Our method effectively addresses the challenges of detecting vehicle targets in wide-area SAR images, and has the potential to significantly enhance real-time reconnaissance tasks and promote the widespread application of remote sensing technology in various fields.

**Keywords:** vehicle detection; SAR imagery; remote sensing images

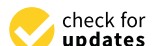



## 1. Introduction

Radar observations play a vital role in today's modern society [1–3]. In particular, Synthetic Aperture Radar (SAR) can achieve high-resolution microwave imaging, which is seldom affected by weather and environment. Furthermore, SAR has great cloud-penetrating capability [4,5]. These advantages over visible optical imaging systems facilitate SAR becoming an ideal detector for high-definition, high-resolution and wide-area imagery [6,7]. In recent years, the demand for remote sensing observations has been steadily increasing, so that there is huge growth of the SAR imagery data. Therefore, current research focuses on how to detect the objects of interest in large amounts of SAR images.

In the past few years, SAR technology has been widely used in the field of remote sensing [8–10], especially in the detection of ground targets [11]. However, with the increase in resolution of SAR satellites, wide-area high-resolution SAR images have generated massive amounts of data, posing challenges for real-time target detection. This is particularly true in the field of military reconnaissance, where there is a high demand for time-sensitive surveillance [12] and reconnaissance. Although the resolution of satellite-borne SAR is able to reach 1 m, detection of small vehicles, such as those 5 m long and 1 m wide, is still of great difficulty. In contrast, detection of large trucks and military vehicles is relatively easier.

On one hand, advanced image processing and analysis algorithms, such as deep-learning-based target detection methods [13], can be used to improve detection performance. Deep learning techniques such as Convolutional Neural Networks (CNNs) have shown strong performance in target detection tasks in recent years. On the other hand, efficient performance is particularly important in airborne SAR systems as it can provide critical information for time-sensitive reconnaissance tasks, thereby providing valuable support for decision-makers. In order to resolve the contradiction between slow processing speed of wide-area SAR images and the demand for fast detection of time-sensitive targets, researchers are dedicated to developing more efficient detection methods. To meet the requirements for efficient performance, existing algorithms need to be optimized to reduce computational complexity and redundant calculations. This can be achieved through targeted design of lightweight neural network architectures or adoption of integrated model frameworks.

In summary, resolving the contradiction between the slow processing speed of wide-area SAR images and the demand for fast detection of time-sensitive targets is of great significance for the development of the remote sensing field. By continuously improving and optimizing detection algorithms to increase the processing speed and target detection performance of SAR images, it is hoped that stronger support can be provided for real-time reconnaissance tasks, further promoting the widespread application of remote sensing technology in both military and civilian fields.

Vehicle target detection in SAR images faces many challenges, including targets of different scales and orientations, complex backgrounds and noise [14,15], and the similarity in texture between vehicle targets and other ground objects due to the SAR imaging mechanism [16]. Firstly, targets of different scales and orientations may exhibit different features and shapes in SAR images, requiring the detection algorithm to have strong scale adaptability. To solve this problem, multi-scale feature extraction and fusion methods can be used to better capture and distinguish targets of different scales. Secondly, complex backgrounds and noise have a significant impact on the accuracy of SAR vehicle target detection. The features of ground objects in complex backgrounds may interfere with vehicle targets, making it difficult for the detection algorithm to accurately identify targets. To reduce this impact, image preprocessing, feature selection and denoising methods can be used to reduce the interference of complex backgrounds and noise on detection results. In addition, due to the characteristics of the SAR imaging mechanism, the texture of vehicle targets and other ground objects in SAR images may be similar, making target detection more difficult. To address this challenge, more robust feature descriptors and deep-learning-based target detection methods can be introduced to improve the ability to distinguish targets from backgrounds. In summary, to address the challenges faced by vehicle target detection in SAR images, researchers need to consider a variety of techniques and methods to improve the accuracy and robustness of detection. By continuously improving and optimizing detection algorithms, it is hoped that efficient and accurate vehicle target detection in SAR images can be achieved under complex background and diverse target conditions.

The traditional method for target detection in SAR images is the CFAR detection method [17–19]. However, there are many limitations to using this method. For example, it assumes that the background clutter follows a Gaussian distribution and requires a significant difference in intensity between the target and the background clutter. These limitations make the CFAR detection method [20] perform well in simple scenarios but perform poorly in SAR image target detection tasks with complex backgrounds.

To address the issue of real-time detection, the YOLO [21–24] and SSD [25] frameworks were developed. They achieved an integrated approach to object detection, eliminating the need to extract regions of interest. All feature extraction, detection and recognition are performed through a single neural network. However, since the target vehicles in SAR images are small, the SSD algorithm does not perform well in detecting small targets. In contrast, the YOLO series of algorithms has been continuously developed in recent years.

YOLOv5 and YOLOX [26] are two object detection models based on the YOLO series, each with its own characteristics, suitable for different application scenarios and requirements. YOLOX is an improvement over YOLOv3, introducing a decoupled head structure, and using the anchor free anchor box method, as well as SimOTA optimal transport and other sample-matching strategies. Its advantages are high performance, and that it can achieve end-to-end detection, and its disadvantage is that the model is larger and requires more computing resources. YOLOv5 is an improvement over YOLOv4 [24], using Focus layer, CSPDarknet53 as the backbone network, SPPF as the neck, SiLU [27] activation function, Mosaic, Copy paste [28] and other data augmentation methods. Its advantages are fast speed, lightweight model and support for multiple backend inference, and its disadvantage is that the detection head has not changed much, as it still uses an anchor-based method. It not only has a more sophisticated network structure but also incorporates many advanced technologies in the field of deep learning, enabling it to surpass many other algorithms in terms of detection accuracy and speed. YOLOv5 and YOLOX have similar performance on the COCO benchmark [29], whereas YOLOv5 has a greater advantage in the trade-off of speed and performance. Therefore, we have chosen YOLOv5 as the basic detection model for vehicle target detection in SAR images under complex scenarios. It employs a convolutional neural network to automatically extract intricate features from complex SAR images, avoiding the difficulty of selecting manual features.

In wide-area SAR image vehicle target detection, a significant issue is the extremely uneven spatial distribution of targets within the SAR image. Most of the area in the image to be tested does not contain vehicle targets, and vehicle targets are often concentrated in scattered small areas. For example, in a wide-area image of $3274 \times 2510$ (as Figure 1), vehicle targets are only concentrated in a very small area. For the wide-area SAR image, sliding-window detection of the image is most common method. For detection methods based on convolutional neural networks (CNNs), a sliding window approach is typically used to perform detection on the entire image. However, due to the uneven distribution of vehicle targets, during target detection, a large amount of computational resources and time will be wasted on detecting areas without targets. This results in a large amount of redundant computation, greatly reducing the computational efficiency of the target detection model. Moreover, it is worth noting that moving vehicles could cause defocusing in SAR images, which could not only severely affect detection performance, but also introduces additional complexities to the detection task due to the need to account for motion blur and potential object distortion. As for wide-area SAR image vehicle target detection, existing datasets [30,31] are mainly focused on static vehicles in remote sensing reconnaissance tasks. Therefore, our approach follows this direction and specifically is designed for stationary vehicles.

To overcome this challenge, we propose a two-stage vehicle target detection method to improve detection efficiency and reduce unnecessary computational overhead. Due to the scattering characteristics of SAR radar and the presence of speckle noise, SAR images may be affected by various noises, which makes the targets easily confused with the background noise. To address this problem, we introduce a new attention mechanism, the CAM module, into the FPN [32] network to effectively fuse the visual features of different scales. By fully utilizing the contextual information, the interference of background noise is suppressed. Meanwhile, since the category prediction, location information and confidence score have different distributions, coupling them in one detection head will lead to slow convergence and performance degradation. As for the classical scenes of the reconnaissance task, the landform of background regions has a very similar visual appearance to vehicle targets in SAR imagery. The original coupled heads from YOLOv5 would be confronted with the difficulty of discriminating the background landform with foreground targets. To resolve this confusion issue, decoupled detection heads derived from YOLOX are adopted to separately predict the category, location and confidence, thus accelerating the model convergence speed and improving the detection accuracy.

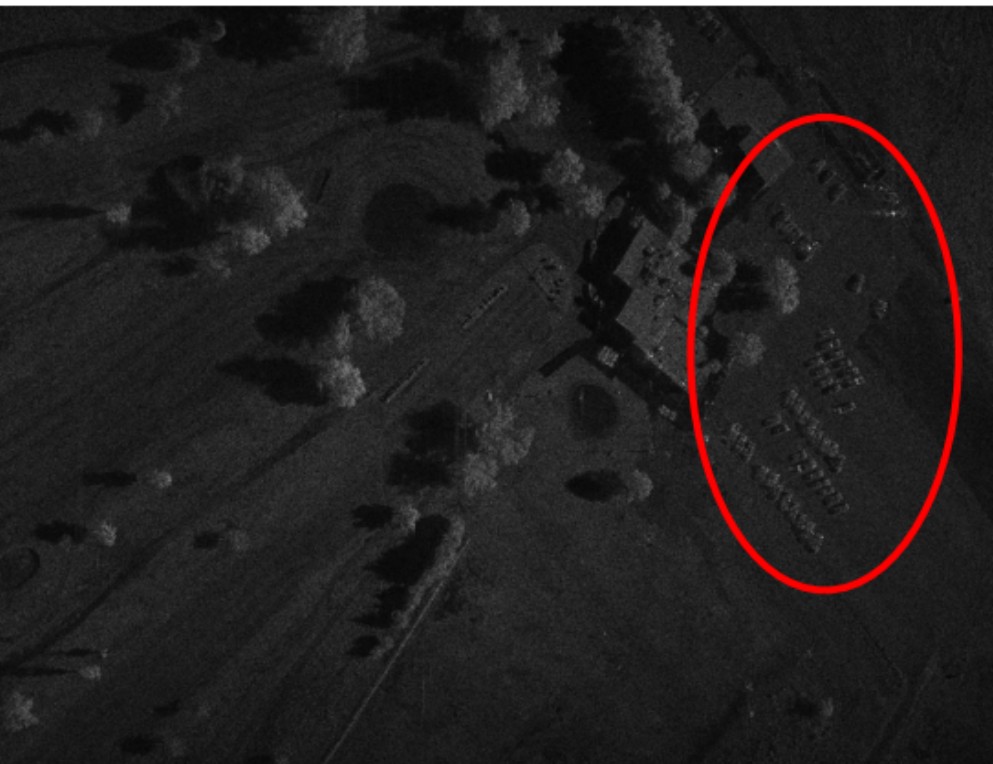

**Figure 1.** Diagram of vehicle target clustering in a SAR image. A red box highlights the region where vehicle targets are concentrated in a localized area.

Considering the uneven spatial distribution of vehicle targets in reconnaissance scenarios, we first use background patches for auxiliary supervision of the detection model to enhance its discrimination ability and recall rate for foreground and background. Secondly, we use our model's foreground–background distinction mechanism to achieve a coarse judgment ability, which is used to optimize the subsequent detection process. By judging the overall category of image patches, we can avoid false positives caused by local noise and only perform detection operations on foreground image patches. On the basis of improving the detection efficiency, the model accuracy is further improved. In addition, the YOLOv5-based detection algorithm requires a large amount of data support to train a model with good detection performance, while there are not as many SAR open datasets as optical images, let alone SAR images containing vehicle targets. Therefore, adding background patches without targets to the training set can make better use of the dataset information, reduce the model's requirement for the number of training samples and improve the model's convergence speed.

## 2. Methods

As for wide-area SAR images, the regions containing vehicle targets only account for a small portion of the entire image, whereas others belong to the background. If a patch category can be determined, it will undoubtedly save computational effort to perform target detection only on the foreground. Therefore, we propose a vehicle detection model based on a coarse-to-fine paradigm, specifically, a coarse-to-fine detection YOLO-based method with CAM Attention, named CF-YOLO. For the input patches, we first determine whether they are foreground or background, and only perform detection on foreground patches.

The proposed coarse-to-fine paradigm still involves using a sliding window to generate patches for subsequent detection, and a certain ratio of overlap is set during the sliding window process. During the detection phase, only foreground patches are sub-

jected to detection, while background patches are not processed further in the subsequent detection operations.

The model structure proposed in this section is shown in Figure 2 and consist of two parts: coarse judgment for foreground and background patches, and fine detection exclusively for positive patches. The coarse judgment branch determines whether the input image is a foreground or background patch. If the input image is a background patch, the detection branch is skipped; if the input image is a foreground patch, it is sent to the detection branch for vehicle target detection. The detection network and classification (judgment) network share a common feature extraction backbone. This backbone sharing can not only reduce the parameters of the whole model, compared to the independent frameworks using two different networks, but also can improve the inference speed if the judgment results can be used to optimize the detection process. Furthermore, the judgment task of fore/background will provide more abundant knowledge to the backbone to resist noise interference. In this structure, the feature extraction backbone adopts the CSPDarkNet structure of YOLOv5.

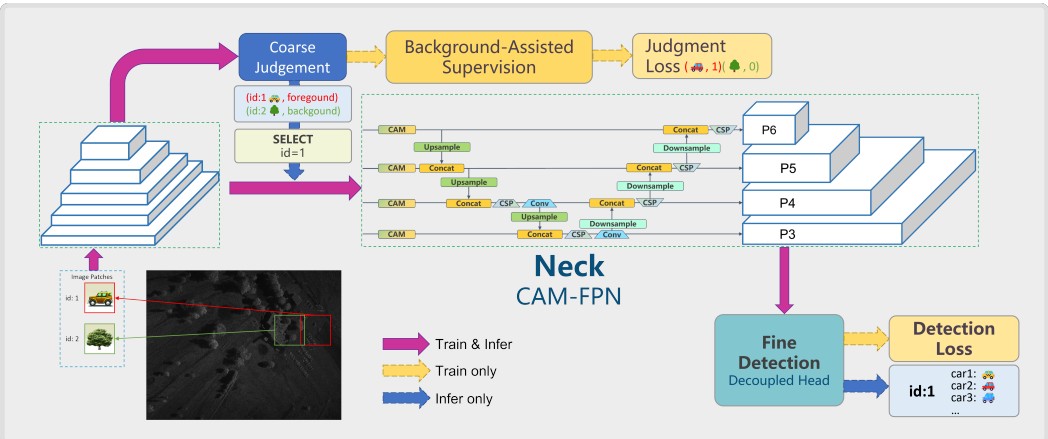

**Figure 2.** Schematic diagram of the integrated coarse-to-fine structure.

## 2.1. Coarse Judgment with Assisted Supervision

Due to the uneven spatial distribution of vehicle targets in reconnaissance SAR images, a foreground–background discrimination ability will facilitate subsequent detection optimization. Specifically, during the inference stage, the model first performs a coarse judgment on the image patches for foreground and background categories. Based on the discrimination results, detection is performed on the foreground image patches, while background image patches do not require detection. By performing an overall judgment on the image patch categories, false alarms and mis-detections caused by local noise in the background image patches can be avoided. In this way, by utilizing the global image's contextual semantics, the model's detection accuracy is further improved while enhancing detection efficiency.

Therefore, we include background images without targets into the detection model's training as an auxiliary supervision for the foreground–background discrimination task, thereby further improving the model's recall rate.

### 2.1.1. Coarse Judgment and Optimization

First, load a large-scale SAR image and use an overlapping sliding window approach to crop it into $448 \times 448$-sized images for input into the model. Then, extract features through the backbone and input the feature maps into the coarse decision branch to determine whether there are targets, obtaining positive patch indices. Use the foreground patch indices obtained from the classification branch to find the corresponding foreground feature maps. Send the corresponding feature maps into the feature map queue. If the number of feature maps in the queue is greater than the predefined value $n_{BS}$ (equivalent

to batch size in model training and inference), then take the first bs feature maps from the queue and send them into the detection branch for detection. Process the detection results appropriately to obtain the corresponding results.

Generally, wide-area images are sliced into smaller pieces using a sliding window approach, resulting in hundreds of smaller images, which is far more than the maximum number of images a GPU can process at once. While during inference, most of the image patches do not contain any targets, the number of remaining images is often very few. After a batch of images goes through the classification branch, only the feature maps of the small number of remaining foreground patches will be sent to the detection branch for detection after each batch of images is classified; this can further improve detection efficiency, so that our detection method can reasonably utilize computational resources.

### 2.1.2. Judgment-Assisted Supervision

For background patch input, the model will still perform forward propagation. When the forward propagation features reach the coarse judgment module, it determines whether the current image has annotation information. If there are box annotations, it is a foreground patch. Otherwise, it is a background patch. We treat this coarse judgment as a classification task of discriminating the foreground from background. This learning process can provide supplementary knowledge to the backbone, as a judgment-assisted supervision. Specifically, the classification loss of the positive and negative within a training batch, $L_{\text{batch}}$ is defined as follows:

$$L_{\text{batch}} = \frac{\sum_{i=1}^{n_{\text{F}}} L_{\text{CE}}(x_i, 1) + \lambda_{\text{back}} \cdot \sum_{j=1}^{n_{\text{B}}} L_{\text{CE}}(x_j, 0)}{n_{\text{BS}}}, \tag{1}$$

where $n_{\text{F}}$ and $n_{\text{B}}$ are the numbers of foreground and background patches respectively; $L_{\text{CE}}$ is the Cross Entropy loss function [33]; $\lambda_{\text{back}}$ is the hyperparameter for balancing foreground and background losses.

### *2.2. CAM and Improved Feature Pyramid Network*

Due to the scattering characteristics of SAR, its imagery may be affected by various types of noises (such as scattering and speckle). Reconnaissance scenes typically contain complex terrain and feature characteristics, and these background objects may generate texture features similar to those of the target vehicles, making it extremely easy for the target features to be confused with background noise.

Attention mechanism, as a mechanism for the brain to process images entering the human eye, plays a significant role in the human visual system. When an image enters the eye, this mechanism allows the visual system to selectively focus on salient regions, ignoring non-essential or targetless areas, essentially concentrating on areas of interest [34]. This mechanism enables the acquisition of more detailed information from the focused regions without wasting effort on useless and distracting information. However, the attention module used in this chapter is different from those introduced earlier. SE modules [35,36] generally use max pooling or average pooling to process channels, which results in the loss of object location information, crucial for capturing target structures in computer vision tasks. CBAM [37] aims to exploit object location information by reducing the channel dimensions of input feature maps, but its use of convolutional operations, which can only capture local relationships for spatial attention, fails to build dependencies between distant pixels in feature maps [38]. While the non-local network addresses CBAM's issue, it usually has a large computational load and is often used in large-scale networks.

### 2.2.1. CAM Attention Layer

To address the issue of target features being easily confused with background noise, we propose a new attention mechanism—the CAM module—and incorporate it into the FPN network to effectively fuse visual features at different scales. By fully utilizing contextual information, the interference of background noise can be suppressed. Therefore, taking into

account the advantages and disadvantages of existing methods, the attention mechanism adopted in this chapter is different from those mentioned above, and its structure is shown in Figure 3.

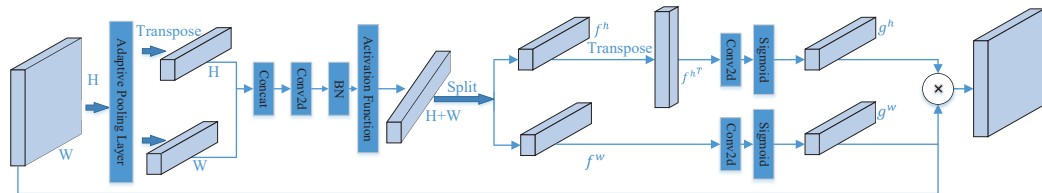

**Figure 3.** CAM Attention Mechanism Structure Diagram.

CAM is aimed at improving the feature representation ability of any intermediate feature map in the network. For a given feature map tensor of an intermediate layer in a convolutional neural network,

$$X_{\text{in}} = [x_1, x_2, x_3, \cdots, x_c] \in R^{C \times H \times W}, \tag{2}$$

CAM can maintain the size of the feature map, ensuring the output and input feature map are identical in size as

$$Y_{\text{out}} = [y_1, y_2, y_3, \cdots, y_c] \in R^{C \times H \times W}. \tag{3}$$

To encode the target location information accurately and capture the attention of the input feature along the height and width directions, CAM applies adaptive global average pooling on the input feature map $X_{\text{in}}$ in both width and height directions, obtaining two feature maps $Z_w$ and $Z_h$. Their dimensions are $C \times 1 \times W$ and $C \times H \times 1$, respectively, indicating the features of $X_{\text{in}}$ along the width and height directions. Among them, the value of the $c$-th channel of $Z_w$ at width $w$ is

$$z_c^w(w) = \frac{1}{H} \sum_{0 \le j < H} x_c(j, w), \tag{4}$$

Similarly, the value at the height of $h$ for the $c$-th channel of $Z_h$ is

$$z_c^h(h) = \frac{1}{W} \sum_{0 \le i < W} x_c(h, i) \tag{5}$$

By applying the above two equations, we obtain a global receptive field and a precise representation of the object positions. These two transformations allow our CAM attention module to capture large-scale dependencies along one spatial direction, while preserving the position information along another spatial direction. This facilitates the network to locate the objects of interest more accurately.

In order to extract the attention map from horizontal and vertical direction separately, firstly, $Z_h$ is transposed to obtain a feature map with a dimension of $C \times 1 \times H$. Then, concatenate it with $Z_w$ along the spatial dimension (height or width) to obtain a feature map $F_1$ with a dimension of $C \times 1 \times (H + W)$. Next, apply a $1 \times 1$ convolutional layer to $F_1$, reducing the feature channels by a factor of $r$. Thus, the dimension of the feature map after convolution becomes $C/r \times 1 \times (H + W)$. Finally, perform batch normalization on the feature map, and then apply a non-linear operation as

$$R(x) = x * (\text{Relu}(x + 3))/6, \tag{6}$$

where $x$ is the input feature map. The overall transformation can be described by the following formula:

$$F = R\Big(BN\Big(\text{Conv2d}_{1\times1}\Big(\big[Z_h{}^T, Z_w\big]\Big)\Big)\Big), \tag{7}$$

where $[*]$ denotes the concatenation operation on the spatial dimension of feature maps; $T$ denotes the transpose operation; Conv2d denotes the convolution operation; BN is the batch normalization operation; $R$ is the nonlinear transformation function.

By applying the F transformation, we obtain the intermediate feature map $f$ with the dimension of $C/r \times 1 \times (H + W)$. Then, we split $f$ along the spatial dimension into two independent feature maps $f^h$ and $f^w$, which have the dimensions of $C/r \times 1 \times H$ and $C/r \times 1 \times W$, respectively. At this point, we need to transpose $f^h$, so that its dimension becomes $C/r \times H \times 1$. Next, we perform the $1 \times 1$ convolution operation on these two feature maps separately, making the channel number of the convolved feature maps the same as that of the input feature map $X$, that is, the dimensions of the convolved feature maps are $C \times H \times 1$ and $C \times 1 \times W$, respectively. Finally, we apply the nonlinear transformation to the two convolved feature maps. Here, we use a simple sigmoid transformation as the nonlinear transformation. The transformed feature maps are denoted as $g^h$ and $g^w$, respectively, and we have the following formula:

$$f^h, f^w = \text{Split}(f), \tag{8}$$

$$g^h = \sigma\Big(\text{Conv2d}_{1\times1}\Big(f^{h^T}\Big)\Big), \tag{9}$$

$$g^w = \sigma(\text{Conv2d}_{1\times1}(f^w)), \tag{10}$$

where Split denotes the operation of splitting $f$ along the spatial dimension into two feature maps, namely, $f^h$ and $f^w$; $\sigma$ denotes the nonlinear transformation sigmoid, which is used to activate the feature maps after convolution.

After the transformations of Equations (4), (5) and (7)–(10), the initial input feature map $X$ obtains two attention weight maps $g^h$ and $g^w$ along the spatial dimension. Then, the output of CAM can be expressed as:

$$Y = X * g^h * g^w, \tag{11}$$

### 2.2.2. CAM-FPN: Improved FPN Network with CAM Module

For CNN-based object detection models, the backbone network extracts visual features from different scales, while the FPN network fuses features from different scales to generate feature maps for targets of various sizes. However, noise from background interference is also mixed in. In order to enable the model to adaptively suppress background noise interference, we incorporate the CAM module into the FPN network. This allows the model to adaptively filter meaningful semantic features and suppress irrelevant noise interference, thereby further enhancing the model's feature extraction capabilities and making it more robust to unrelated noise.

The insertion position of the CAM attention module in the network, as described above, is shown in Figure 4. For the last feature map C5 generated by the feature extraction module, it passes through the CAM module to obtain the feature map M51, and simultaneously goes through a convolution operation to obtain the feature map M52. The corresponding elements of M51 and M52 are added together to obtain the fused high-level feature map M5.

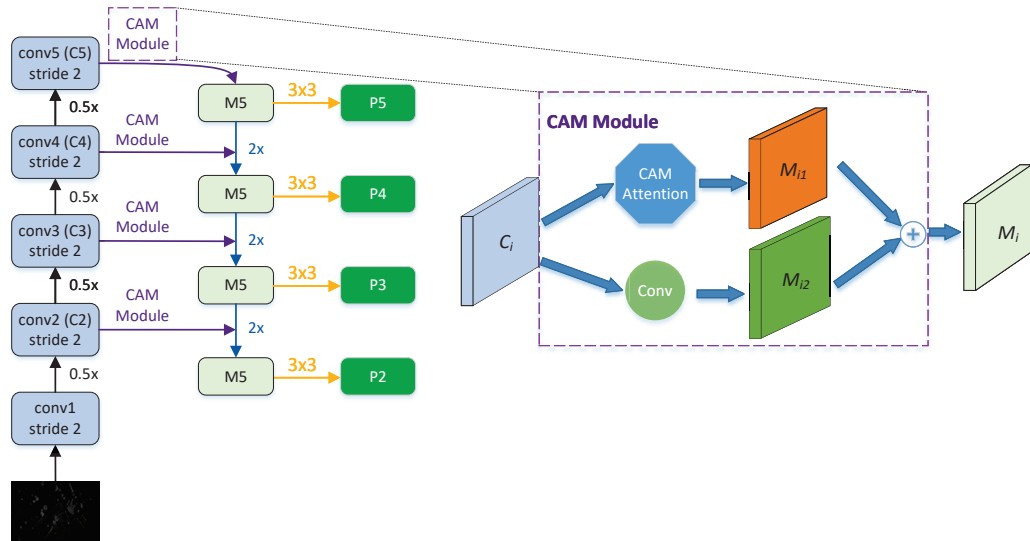

**Figure 4.** Schematic diagram of the attention mechanism insertion position in CAM module from C5 to M5. CAM module consists of a CAM attention, Conv layer and adding operation.

### 2.3. Decoupled Detection Head

In traditional detection heads, the output channel number is changed for each feature layer through convolution, making the output channel number ($n_a \times (1 + 4 + n_c)$), where na represents the preset anchor boxes per point, 1 indicates the predicted target confidence, 4 represents the four coordinate offsets (center point xy, height-width wh), and nc represents the probability for each category. This traditional approach completes the classification and localization tasks through a single convolution.

However, classification and localization tasks have different focus points when it comes to the features extracted by the backbone. Classification is more concerned with how similar the extracted features are to existing categories [39]. Localization, on the other hand, focuses more on the deviation from the ground truth box (GTBox) coordinates in order to adjust the bounding box parameters. Therefore, completing both localization and recognition tasks on the same feature map might not yield good results [40].

Specifically, in order to have distinct distributions for category prediction, location information and confidence scores, coupling them in a single detection head may lead to slow convergence and reduced performance due to inconsistent distributions. Here, we adopt the approach used in YOLOX, which decouples the detection head to separately predict categories, locations, and confidence scores, thereby accelerating model convergence and improving detection accuracy.

Therefore, in this section, we choose the decoupled detection head, the structure of which is shown in Figure 5.

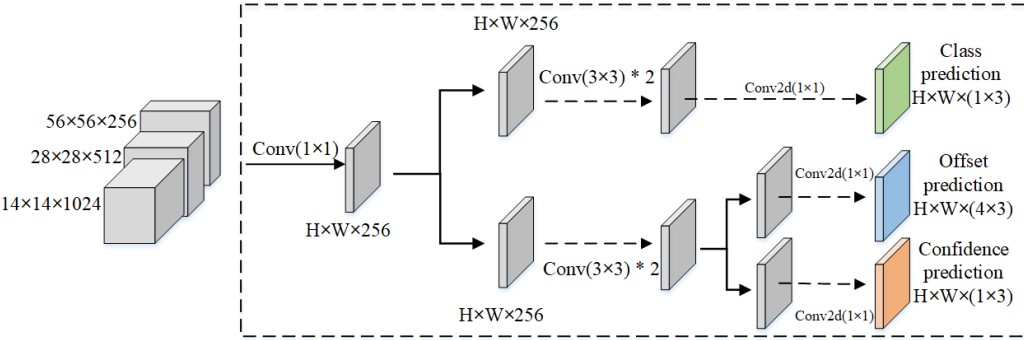

**Figure 5.** Decoupled Head Structure Diagram.

## 3. Experiment

### 3.1. Dataset

#### 3.1.1. MiniSAR Dataset

In 2006, Sandia National Laboratories released a measured dataset of SAR vehicle targets to the public. This dataset, known as MiniSAR, was derived from raw data using a specific algorithm. The dataset comprises 20 images featuring complex backgrounds such as roads, buildings, vegetation and so on. Out of these images, 17 contain various types and sizes of vehicle targets, while three are devoid of any vehicle targets. The images are either 1638 × 2510 or 3274 × 2510 pixels in size, with a spatial resolution of 0.1 m. The training set consists of fourteen images with vehicle targets, and the test set consists of six images (three of which without targets). Two examples of MiniSAR images are shown in Figure 6. Figure 6a shows an image example of the dense building area in the MiniSAR Dataset, where many vehicle targets are parked in clusters. Figure 6b shows an image example of the wide road area, where trees, buildings and vehicle targets are scattered in the image.

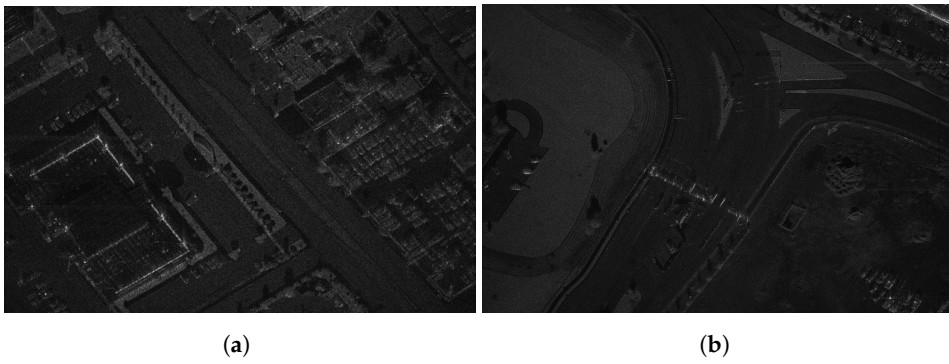

(**a**) (**b**)

**Figure 6.** Examples of MiniSAR images. (**a**) Dense building area. (**b**) Wide road area.

#### 3.1.2. MSTAR Dataset

The MSTAR dataset [30] is divided into two categories: target dataset, with each image having a simple background and a single small vehicle at the center; and clutter dataset, with each image featuring no vehicles and diverse scenes, and being larger in size.

The target dataset contains multiple stationary vehicles in a military base, taken at various angles and orientations. These images are SAR slices, having relatively uniform backgrounds. Each pixel corresponds to 0.3 m, and most images are 128 × 128 in size, with the exception of ZIL131 vehicle slices that measure 192 × 193. The raw data are complex-valued data with SAR information, and a specific algorithm is required to extract pixel information from them and obtain image data. We use 10 different models of vehicle targets, as shown in Table 1.

**Table 1.** Vehicle type comparison table.

| D7 | BTR60 | 2S1 | T62 | BRDM2 | BMP2 | ZSU234 | ZIL131 | T72 | BTR70 |
|---|---|---|---|---|---|---|---|---|---|
| Bulldozer | Armored car 1 | Howitzer car | Tank 1 | Armored car 2 | Infantry fighting vehicle | Anti-aircraft vehicle | Military truck | Tank 2 | Armored car 3 |

Figure 7 showcases SAR image examples of 10 distinct vehicle targets, while Figure 8 displays the corresponding optical images. Within the MSTAR target dataset, empirical data with depression angles of 15° and 17° are included. For the purpose of this research, we opted for the data with a 17° depression angle as training samples, and the ones with a 15° depression angle as testing samples. Importantly, although T72 and BMP2 each encompass three variants, we only selected the T72 variant SN_132 and the BMP2 variant SN_9563 for the training and testing sets.

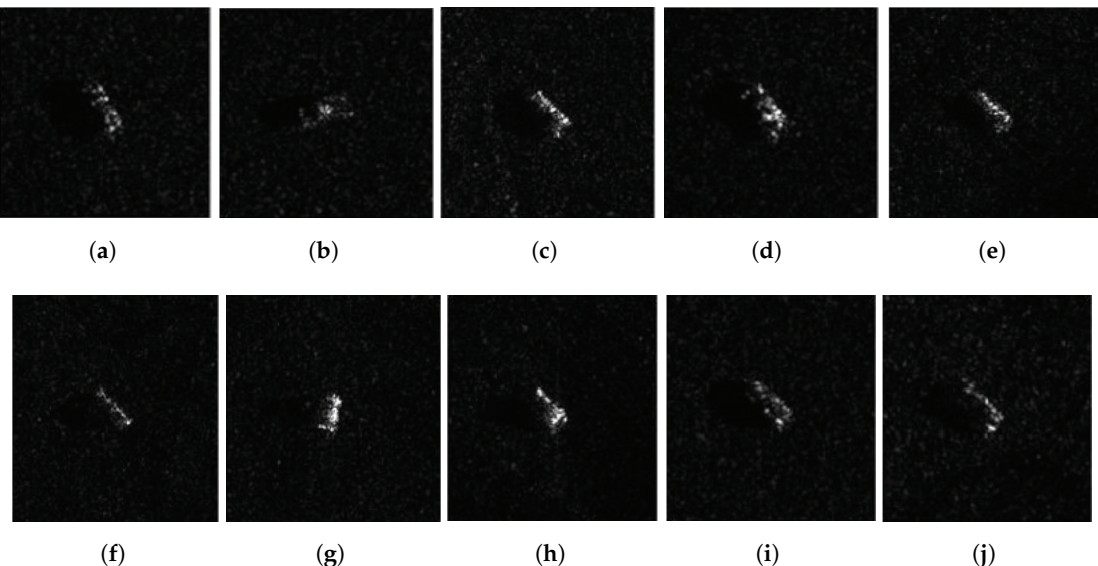

**Figure 7.** SAR images corresponding to 10 classes of vehicle targets. (**a**) BTR70. (**b**) BMP2. (**c**) 2S1. (**d**) T62. (**e**) T72. (**f**) ZIL131. (**g**) D7. (**h**) ZSU234. (**i**) BTR60. (**j**) BRDM2.

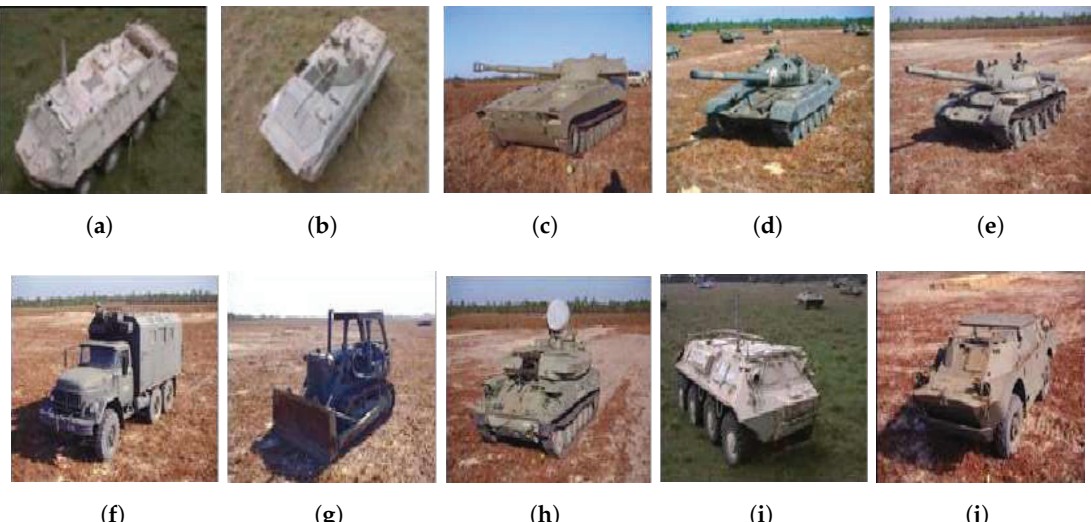

**Figure 8.** Optical images corresponding to 10 classes of vehicle targets. (**a**) BTR70. (**b**) BMP2. (**c**) 2S1. (**d**) T62. (**e**) T72. (**f**) ZIL131. (**g**) D7. (**h**) ZSU234. (**i**) BTR60. (**j**) BRDM2.

### 3.1.3. Wide-Area SAR Vehicle Detection Dataset, WSVD

To carry out the task of vehicle detection, we need wide-area SAR images that contain multiple types of vehicles. However, existing SAR image datasets either have simple backgrounds (such as the MSTAR target dataset) or do not include target vehicles (such as the clutter dataset). Therefore, we embed the vehicle targets from the MSTAR target dataset into background clutter images with diverse scenes, creating new SAR images for training and testing. These created images have varied scenes that include different non-target areas such as shrubs, fields, buildings, lakes, etc.

To evaluate the performance of our proposed algorithm, we developed a novel dataset, called Wide-area SAR Vehicle Detection Dataset (WSVD), for vehicle detection in wide-area SAR images.

(1)  Target and background fusion. To simplify program processing, we first annotated each vehicle target image using annotation software, obtaining a series of XML files that record the position information of each vehicle target image. Next, based on the insertion point coordinates in the background image, we could quickly calculate the

new annotation information of the vehicle target in the background image. However, not all of the 100 clutter background images were appropriate for inserting vehicle targets, such as the two images shown in Figure 9, since it is known that a car could not appear on a tree or building. Figure 9a displays a scene of a forest area, densely blanketed with vegetation, leaving almost no visible ground surface. Figure 9b illustrates a compact region featuring a mix of structures and greenery. The structures sit tightly clustered, with meager gaps, diminishing the probability of accommodating vehicle targets. To prevent vehicles from appearing in unreasonable positions, we manually selected the insertion points in each image.

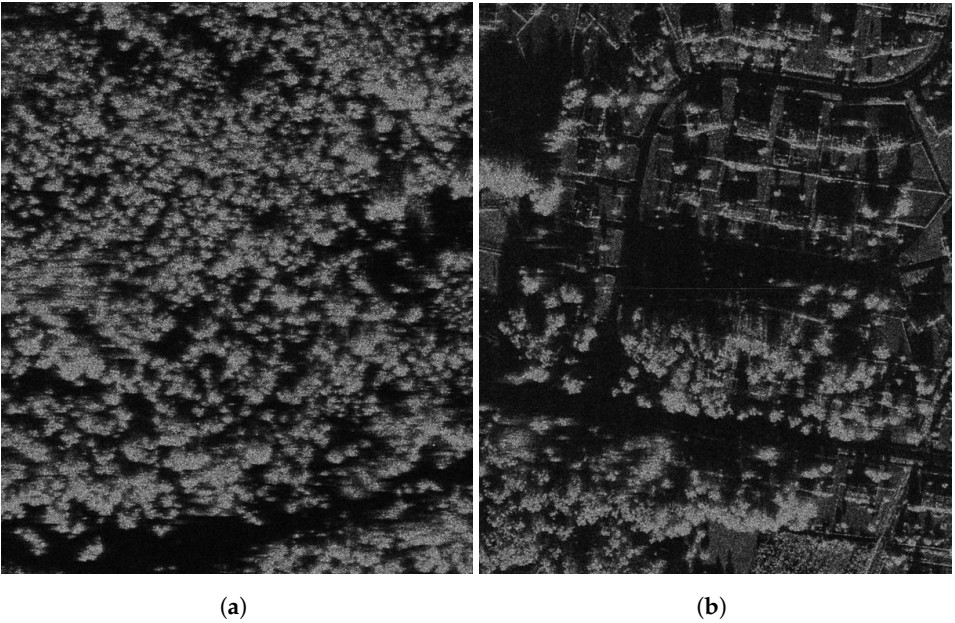

(**a**)  (**b**)

**Figure 9.** Examples of images lacking proper insertion points. (**a**) Dense vegetation area with trees. (**b**) Dense urban area with buildings.

The following steps describe the method of fusing and generating vehicle target images with complex background clutter images:

(a) Assume that $A$ is the target image, where the vehicle target's bounding box is defined by the coordinates of the upper left corner $(x_{Amin}, y_{Amin})$ and the lower right corner $(x_{Amax}, y_{Amax})$. Let $B$ be the complex background image, and $(x_0, y_0)$ be the insertion point manually chosen in $B$.

(b) First, align the upper left corner of the target image $A$ to the insertion point $(x_0, y_0)$, and then find a region $B_1$ in the background image $B$ that matches the size of $A$. Next, convert both $A$ and $B_1$ to HSV space, where $V$ represents the image brightness, and compute the mean values of $A$ and $B_1$ on the $V$ channel, denoted as $avg_A$ and $avg_B$, respectively. Then, multiply the $V$ channel values of $A$ in HSV space by $\frac{avg_B}{avg_A}$, obtaining the adjusted image $A_1$. Finally, convert $A_1$ from HSV space back to BGR space, resulting in the final image $A_2$.

(c) Substitute $B_1$ with $A_2$, thus successfully embedding the target image $A$ into the background image $B$.

(2) Train and test set creation. From 100 complex background images, we randomly selected 75 for the training set and 23 for the test set (excluding two background images without suitable insertion points). We composited 2747 vehicle target images at 2756 manually selected insertion points in the 75 background images using the previously described method. Since the synthesized wide-area SAR images containing various military vehicle targets were too large for direct training, we applied the same cropping method as used for MiniSAR images. After cropping, we generated 1194 initial training images as DATASET_INIT. We then used a static optical expansion

method based on image augmentation to process the SAR images in DATASET_INIT, creating corresponding expanded training samples. We selected 73 images for each vehicle category from the test vehicle target images, totaling 730 vehicle target images, and randomly pasted them onto the 727 insertion points in the 23 test background images. Ultimately, we constructed 23 test images. The numbers of training and test sets are provided in Table 2. Figure 10 displays two of the synthesized images. Figure 10a depicts the simulation results of fusing vehicle targets in a wide-area and flat terrain area, where multiple vehicle targets are uniformly distributed on the ground. Figure 10b demonstrates the simulation results of adding vehicle targets in a dense area with trees, vegetation and buildings, where only the regions with adequate space for vehicle parking are selected to fuse simulated vehicle targets, due to the occurrence of some areas covered by dense vegetation.

**Table 2.** Number of training and testing images.

|  | Synthesized | Cropped Patches | Expansion Ratio | Expanded Patches |
|---|---|---|---|---|
| Train Set | 75 | 1194 | 10 | 13,134 |
| Test Set | 23 | no cropping | - | - |

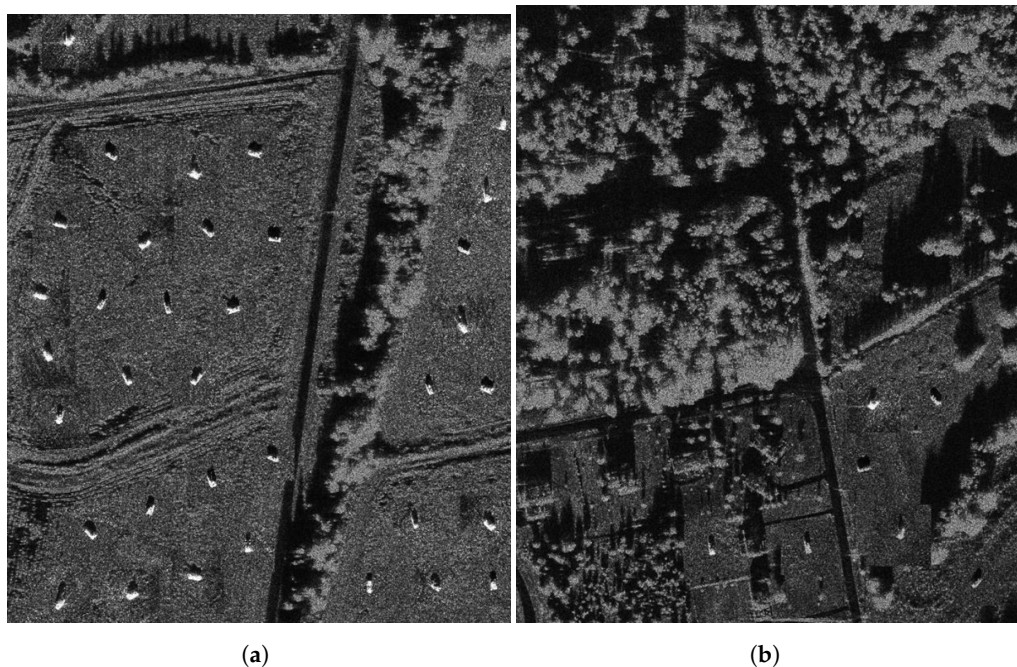

(**a**)          (**b**)

**Figure 10.** Examples of synthesized images of vehicle targets. (**a**) Simulated image in wide flat area. (**b**) Simulated image in dense vegetation and urban area.

### 3.2. Experiments on CF-YOLO

In the proposed improved approach of CF-YOLO, we introduced three enhancements, which are CAM-FPN, judgment-assisted supervision and decoupled detection head. The ablation study results are shown in Table 3, where AP refers to the Average Precision score [41]. As can be seen from the results, each improvement module significantly enhances the performance of the baseline model, with CAM-FPN being the most prominent, achieving an increase of 6.67%.

In order to verify the effectiveness of the proposed module combinations in this study, we used YOLOv5 as the baseline model and added each proposed module to it individually. The experimental results are shown in Table 3.

**Table 3.** Experimental results of various module combinations, where C stands for CAM-FPN, D for Decoupled head and AS for background Auxiliary Supervision.

| Experimental Model | mAP (%) |
| --- | --- |
| YOLOv5 (Base) | 91.69 |
| Base + CAM-FPN (C) | 93.87 |
| Base + Decoupled head (D) | 92.51 |
| Base + Auxiliary Supervision (AS) | 92.86 |
| Base + C + D | 94.82 |
| Base + C + D + AS | 95.02 |
| Base + C + D + AS + Judgment Optimization | 95.55 |

### 3.3. Experiments for the Coarse Judgment Branch

The purpose of this experiment is to evaluate the capability of the coarse judgment branch to discriminate between foreground and background patch, and to examine the effect of the classification threshold on the detection performance. The classification threshold is a criterion that classifies a patch as either foreground or background. The selection of the classification threshold influences the quantity and quality of the feature maps fed into the detection branch, which in turn affects the detection accuracy, recall and time consumption of the detection branch.

Firstly, we compared the classification performance under various classification thresholds on the validation set. The results are presented in Table 4.

**Table 4.** Comparison of classification performance on validation set with different classification thresholds.

| Threshold | Accuracy |
| --- | --- |
| 0.28 | 0.9571 |
| 0.4 | 0.9134 |
| 0.5 | 0.7441 |

As shown in Table 4, with a lower threshold, the classification module improves its ability of identifying foreground, but some foreground patches are still misclassified as background. This suggests that although the coarse judgment module can effectively distinguish most patches, there are still some challenging images that are hard to judge.

Secondly, we compared the judgment performance of the detection branch under different classification thresholds on the test set. The results are given in Figure 11.

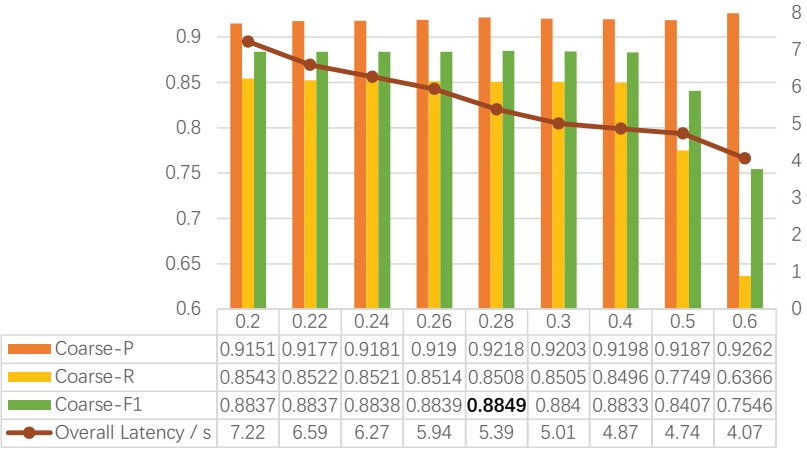

| | 0.2 | 0.22 | 0.24 | 0.26 | 0.28 | 0.3 | 0.4 | 0.5 | 0.6 |
| --- | --- | --- | --- | --- | --- | --- | --- | --- | --- |
| Coarse-P | 0.9151 | 0.9177 | 0.9181 | 0.919 | 0.9218 | 0.9203 | 0.9198 | 0.9187 | 0.9262 |
| Coarse-R | 0.8543 | 0.8522 | 0.8521 | 0.8514 | 0.8508 | 0.8505 | 0.8496 | 0.7749 | 0.6366 |
| Coarse-F1 | 0.8837 | 0.8837 | 0.8838 | 0.8839 | **0.8849** | 0.884 | 0.8833 | 0.8407 | 0.7546 |
| Overall Latency / s | 7.22 | 6.59 | 6.27 | 5.94 | 5.39 | 5.01 | 4.87 | 4.74 | 4.07 |

**Figure 11.** Comparison of judgment performance under different classification thresholds.

As can be seen from the table in Figure 11, with a higher threshold, the classification recall decreases, while the detection time t correspondingly reduces; on the contrary, with a

lower threshold, the recall increases, while the detection latency correspondingly rises. This is because a higher threshold results in fewer feature maps sent to the detection branch, among which some foreground patches containing targets are misclassified as background and missed by the detection branch, leading to a lower recall; while a lower threshold results in more feature maps sent to the detection branch, among which some background patches are misclassified as foreground and cause the invalid computation, leading to longer detection latency.

### 3.4. Comparison Experiments for Detection Optimization

Table 5 presents the comparison results of the proposed model w/o detection optimization via results of coarse judgment in terms of parameter size, inference time, detection accuracy and recall rate. In the partition-based classification model, the classification threshold is set to 0.28. Both models employ the WASD dataset and follow a same setting.

**Table 5.** Comparison results of using judgment optimization.

| Model | Number of Parameters | Overall Latency (s) | mAP (%) |
|---|---|---|---|
| Imp: baseline + C + D + AS | 46,186,759 | 13.22 | 95.02 |
| Imp. + Judgment Optimization | 53,651,143 | 5.39 | 95.55 |

According to Table 5, the partition-based integrated detection model in this section slightly increases in parameter size but reduces inference time and marginally improves the mAP score. The increase in model parameter size mainly stems from the classification branch. In contrast to the classification branch, the detection branch adopts the FPN structure and further adds a PAN structure [42] and detection head. For foreground images, they need to go through both coarse judgment and detection branches for inference, which adds extra computation time. However, since a large number of correctly classified background partitions do not go through the detection branch, the overall inference time is actually reduced.

### 3.5. Performance Comparison with Other Algorithms

The performance of our proposed CF-YOLO model is compared with other algorithms in terms of mAP and detection latency, as detailed in Table 6.

**Table 6.** Comparison of detection performance with other classical algorithms. The configuration of computing resources is GTX 2080Ti graphics card, with the PyTorch programming framework and in the FP32 data type.

| Method | mAP (%) | Overall Latency (s) |
|---|---|---|
| FasterRCNN [43] | 87.86 | 17.71 |
| RetinaNet | 91.60 | 15.37 |
| CF-YOLO | 95.55 | 5.39 |

As shown in Table 6, the CF-YOLO model achieved the highest mAP score, and also had the fastest inference time (5.3871 s). In contrast, the RetinaNet model using FocalLoss function took 15.3725 s to infer 23 test images, which was about three times longer than the CF-YOLO model. Therefore, our proposed model effectively enhanced the detection performance by incorporating attention mechanism and decoupling heads.

To show the superiority of our method more clearly, we compare the detection and recognition results of the three methods in Figure 12. Figure 12a displays the original sub-image with the ground truth boxes corresponding to vehicle targets. As shown in the figure, FasterR-CNN (Figure 12b) has four recognition errors, of which three are caused by inaccurate localization (with IoU less than 0.7) leading to missed detections; RetinaNet (Figure 12c) has three recognition errors and two missed detections; while our method

(Figure 12d) only has two misses due to inaccurate localization, with no recognition errors. This suggests that in the case of a lower IoU threshold of mAP, our method can achieve effective detection and recognition of this sub-image.

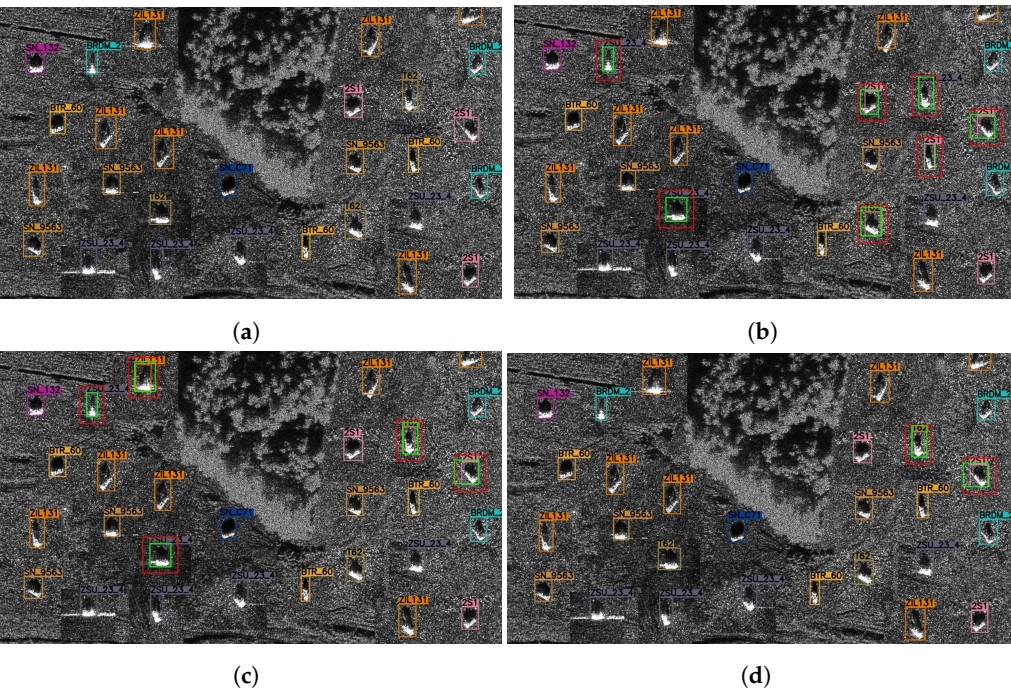

(**a**)

(**b**)

(**c**)

(**d**)

**Figure 12.** Ground truth and detection results of three Methods. Different vehicle types are denoted with color-coded boxes. Red boxes indicate false detections, whereas missed or misclassified objects are represented by green boxes, signifying undetected ground truth boxes. (**a**) Sub-image patch with ground truth. (**b**) FasterRCNN. (**c**) RetinaNet. (**d**) Our method.

*3.6. Supplementary Experiments of Post-Processing of Bounding Boxes*

In practical vehicle detection tasks, relying solely on the Non-Maximum Suppression (NMS) algorithm for post-processing may not yield satisfactory results. The NMS algorithm works by eliminating predicted bounding boxes that have a lower confidence level and an Intersection over Union (IoU) value exceeding a certain threshold with other predicted boxes of the same object. However, this approach can sometimes lead to a "large-box-encloses-small-box" scenario, where a larger predicted box entirely encompasses a smaller one, as illustrated by the red and blue ellipses in Figure **??**.

In the detection results, some boxes only encase a portion of the target, making them relatively small in area. Others successfully encompass the entire target, resulting in a larger area. When both types of boxes represent the same target and their intersection over union (IoU) is below a pre-set threshold, both boxes are retained. However, in practice, the smaller box should also be eliminated, a task beyond the capability of traditional non-maximum suppression (NMS) algorithms. To address this issue, we propose the following approach: For detection boxes of the same output category, we first calculate the intersection area of two boxes, followed by the area of each individual box. Subsequently, we calculate the ratio of the intersection area to each box's area. Boxes for which this ratio exceeds a pre-set threshold $t_{\text{Thres}}$ are eliminated. In other words, any box satisfying the following condition will be discarded:

$$\frac{S(P_{\text{box1}} \cap P_{\text{box2}})}{min(S(P_{\text{box1}}), S(P_{\text{box2}}))} > t_{\text{Thres}} \tag{12}$$

Here, $S$ denotes the operation to calculate the area and $P$ denotes the pixel set of the box. By employing this method, we can successfully remove the smaller box among the two. The results of post-processing are demonstrated in Figure 14. It is noticeable that the

condition of larger boxes containing smaller ones, which is present in Figure **??**, has been eliminated in Figure 14.

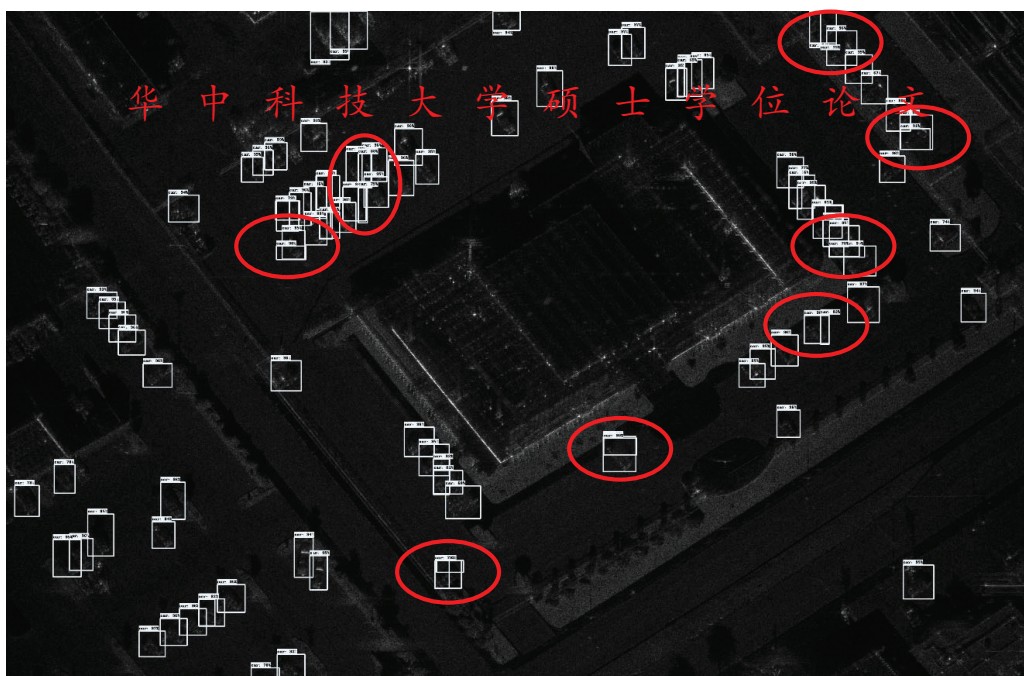

**Figure 13.** Without post-processing, larger predicted boxes may encompass smaller ones, as illustrated by the red boxes in the figure.

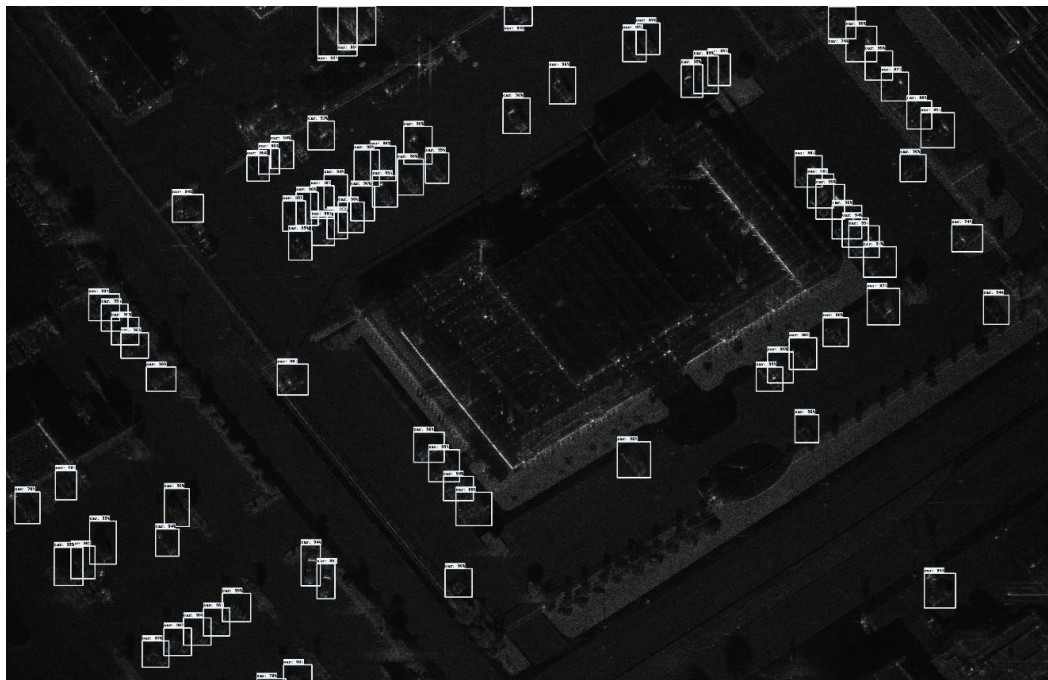

**Figure 14.** Detection result with the improved post-processing.

### 3.7. Discussion

According to the experimental results, the proposed model has excellent performance in vehicle target detection and recognition in SAR images. The method adopts the CAM-FPN feature fusion structure, which can better select the feature information needed by the target at the current scale through the attention mechanism, thereby promoting the

information fusion of high-level and low-level features. In addition, the model also uses a decoupled detection head module, which can make the model more accurate in target localization and classification.

The dataset used in this paper is generated by fusing the target and clutter background image data with a simulation algorithm. Since the simulation cannot fully reproduce the images collected in the real environment, there are some differences between the synthetic region and the surrounding environment, as there are apparent boundary artifacts around the first ZSU234 target vehicle shown in the lower left corner of Figure 12d. This might be utilized by the model as illusory clues for location task. Furthermore, the data amount of 10 vehicle targets in MSTAR is relatively small, so that if the classification algorithm is strong enough to overfit the data, the classification accuracy of 10 vehicle targets in MSTAR data set is possible to reach more than 99%. Therefore, using such a data set for wide-area vehicle target detection and recognition in SAR images result in relatively better performance, while a dataset collected in the real environment can be more challenging.

## 4. Conclusions

In this paper, we propose a vehicle target detection model in the coarse-to-fine paradigm, targeting wide-area SAR images. Firstly, the coarse judgment module performs foreground and background classification on the sliding windowed patch, and then the fine detection branch only conducts forward operations on the feature maps of foreground. This can avoid wasteful detection of non-target areas and improve processing speed. Then, we build the detection model with proposed the CAM attention module to improve the model's fine-grained recognition ability for vehicle targets. Furthermore, we use a decoupled detection head to improve the model's localization ability for targets. Our framework also utilizes background area information and enhances the model's detection performance by supervising the model's judgment loss for the classification of foreground and background patches. We also used a data synthesis method to construct a wide-area SAR image dataset containing 10 types of vehicle targets and uses sliding window overlapping cropping and data augmentation methods to construct a training dataset containing more than 13,000 images. Through comparative experiments, we compared the detection and recognition performance of models with different modules, and benchmarked our algorithm with other typical end-to-end models, demonstrating its superiority. Extensive experimental results show that our method effectively improves detection accuracy and speed, and verifies the feasibility and efficiency of the model.

When determining the presence of targets in partitioned SAR image patches, we find that the first few layers of the backbone network, such as the first three layers, may be sufficient for this task. Therefore, in future work, we plan to further optimize the coarse detection branch of the target detection model by connecting it to the third layer of the backbone network, reducing computation and increasing inference speed. Additionally, we will investigate new feature extraction techniques and attention mechanisms to address complex SAR image scenes and further improve detection accuracy and speed. We will also explore the application of this target detection method in other remote sensing fields, such as ocean monitoring, forestry and agriculture.

**Author Contributions:** Conceptualization, Y.S. and Q.L.; methodology, Y.S. and S.W.; software, Q.L.; validation, H.M., R.F. and T.T.; formal analysis, T.T. and Y.S.; investigation, Y.S.; resources, R.F.; data curation, Q.L.; writing—original draft preparation, Y.S. and Q.L.; writing—review and editing, Y.S. and T.T.; visualization, Q.L.; supervision, J.T.; project administration, J.T.; funding acquisition, T.T. All authors have read and agreed to the published version of the manuscript.

**Funding:** This research was funded by National Natural Science Foundation of China, grant number 42071339, and National Key Laboratory Fund, grant number 6142113210310.

**Data Availability Statement:** The MiniSAR and MSTAR datasets, which are publicly available, were used in this study. The MiniSAR dataset can be accessed at https://www.sandia.gov/radar/pathfinder-radar-isr-and-synthetic-aperture-radar-sar-systems/complex-data/, and the MSTAR dataset can be accessed at [30]. Additionally, new data were generated through simulation methods for this study. However, these data are not publicly available due to data licensing constraints. For further inquiries or requests regarding the data, please contact the corresponding author.

**Acknowledgments:** The authors would like to express thanks for the support from the China University of Geosciences, which we gratefully appreciate.

**Conflicts of Interest:** The authors declare no conflict of interest.

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
