# Peer review of "Vehicle Target Detection Method for Wide-Area SAR Images Based on Coarse-Grained Judgment and Fine-Grained Detection"

_remotesensing, doi:10.3390/rs15133242_

Round 1

Reviewer 1 Report

1)    The detection object is static vehicle targets, which should be clearly stated in the text. If it is a moving vehicle, it will cause defocusing in SAR images,which will seriously affect performance.

2)    There is a lack of detailed explanation on the theoretical model of CAM and improved feature pyramid network. It is recommended to enhance it and theoretically analyze its performance and efficiency.

3)    In Table 6, the evaluation of overall latency should provide the configuration of computing resources.

4)     The dataset used in this paper is generated by fusing the target and clutter background image data with a simulation algorithm. How is the training data of the model generated?

1)    The detection object is static vehicle targets, which should be clearly stated in the text. If it is a moving vehicle, it will cause defocusing in SAR images,which will seriously affect performance.

2)    There is a lack of detailed explanation on the theoretical model of CAM and improved feature pyramid network. It is recommended to enhance it and theoretically analyze its performance and efficiency.

3)    In Table 6, the evaluation of overall latency should provide the configuration of computing resources.

4)     The dataset used in this paper is generated by fusing the target and clutter background image data with a simulation algorithm. How is the training data of the model generated?

The English writing of this paper is generally acceptable. Minor editing of English language is required

Reviewer 2 Report

 The article concerns detecting vehicle targets in wide-area SAR images.The article is contained in a given field. It is relevant and fills a gap in the field.The article describes detecting vehicle targets in wide-area SAR images, and has the potential to significantly enhance real-time reconnaissance tasks.
The conclusions are consistent with the research and respond to the theses presented in the article.The scope of the literature is adequate.

The article is interesting and worth considering for publication. The article describes methods of target detection using SAR technology.

Comments:

1. Please explain whether the pictures in Figures 1 and 6 are the same?

2. Please explain Figures 4 and 5 in the text.

3. Please add a and b to Figures 6, 9 and 10 and describe what they represent.

4. Please refer to figure 12a, b, c and d in the text.

After making changes, the article is ready for publication.
